# Bioactive Profile of Distilled Solid By-Products of Rosemary, Greek Sage and Spearmint as Affected by Distillation Methods

**DOI:** 10.3390/molecules27249058

**Published:** 2022-12-19

**Authors:** Stamatia Christaki, Elisavet Bouloumpasi, Eleni Lalidou, Paschalina Chatzopoulou, Maria Irakli

**Affiliations:** 1Hellenic Agricultural Organization—Dimitra, Plant Breeding and Genetic Resources Institute, 57001 Thessaloniki, Greece; 2Department of Food Science and Technology, School of Agriculture, Aristotle University of Thessaloniki, 54124 Thessaloniki, Greece

**Keywords:** hydro-distillation, steam distillation, microwave-assisted hydro-distillation, distilled solid residues, antioxidant activity, LC-MS, phenolic compounds, identification

## Abstract

By-products of essential oils (EOs) in the industry represent an exploitable material for natural and safe antioxidant production. One representative group of such by-products is distilled solid residues, whose composition is properly modulated by the distillation method applied for the recovery of EOs. Recently, in terms of Green Chemistry principles, conventional extraction and distillation processes are considered outdated and tend to be replaced by more environmentally friendly ones. In the present study, microwave-assisted hydro-distillation (MAHD) was employed as a novel and green method for the recovery of EOs from three aromatic plants (rosemary, Greek sage and spearmint). The method was compared to conventional ones, hydro-distillation (HD) and steam-distillation (SD), in terms of phytochemical composition of distilled solid residues, which was estimated by spectrophotometric and chromatographic methods. Total phenolic content (TPC), total flavonoid content (TFC) and antioxidant activity (ABTS, DPPH and FRAP) results highlighted the distilled solid residues as good sources of antioxidants. Moreover, higher antioxidant activity was achieved for MAHD extracts of solid residues in comparison to HD and SD extracts. A metabolomics approach was carried out on the methanolic extracts of solid residues obtained by different distillation methods using LC-MS analysis followed by multivariate data analysis. A total of 29 specialized metabolites were detected, and 26 of them were identified and quantified, presenting a similar phenolic profile among different treatments, whereas differences were observed among different species. Rosmarinic acid was the most abundant phenolic compound in all extracts, being higher in MAHD extracts. In rosemary and Greek sage extracts, carnosol and carnosic acid were quantified in significant amounts, while trimers and tetramers of caffeic acid (salvianolic acids isomers) were identified and quantified in spearmint extracts, being higher in MAHD extracts. The obtained results pointed out that MAHD extracts of distilled solid by-products could be a good source of bioactives with potential application in the food, pharmaceutical and cosmetic industries, contributing to the circular economy.

## 1. Introduction

Essential oils (EOs) have been used for centuries as medicines, cosmetics and in food preparations. Currently, there is an increasing interest in plant-based ingredients, as they are considered safer compared to synthetic chemicals. The global market for EOs is growing rapidly, due to their multiple uses in various industrial sectors, such as in the food and beverage industry, cosmetics, flavors and fragrances, pharmaceuticals, feed additives, green chemicals in agriculture etc. EOs exhibit antimicrobial and antioxidant properties, and therefore they have been extensively studied as potential antimicrobial agents and natural food preservatives in various food products [1,2,3,4]. Since these properties are directly linked to the EOs’ composition, the method of extraction from the raw material is of key importance for the recovery of EOs of a desired constitution. According to the European Pharmacopoeia, EOs are odorous products, usually of complex composition, obtained from a botanically defined raw vegetable material by hydro-distillation (HD), steam distillation (SD), or a suitable mechanical process [5].

Among the conventional methods, HD and SD are commonly employed for EO recovery from aromatic and medicinal plants [6]. SD is a classic technique widely used by the industry to obtain EOs, whereas HD is mostly applied in laboratory-scale distillations and in specific plant materials: flowers, rose oil, etc. Through the last years, in light of the climate crisis and the search for sustainable green processes, novel distillation and/or extraction methods have been developed, aiming to replace the conventional ones in terms of time, energy and solvent consumption reduction [7,8].

Although the majority of commercially available EOs are obtained through SD and/or HD, these methods present several disadvantages, as reported by many studies in the literature, namely high energy and water consumption, losses and thermal degradation of volatile compounds and long distillation times [9,10]. Therefore, alternative, more promising “green” methods have been studied for the replacement, when feasible, of the conventional distillation processes. 

One of the innovative distillation methods, in terms of sustainability, is microwave-assisted distillation (MAD). More specifically, microwaves, a type of electromagnetic wave, promote the rapid rotation of polar molecules in an electric field through dipole-dipole rotation or ion conduction, enabling the disruption of hydrogen bonds in the extraction system [11,12]. This mechanism leads to the destruction of the plant microstructure and increased solvent absorption in the material, where both heat and mass transfer are performed simultaneously. As a result, MAD is characterized by short distillation times compared to HD and SD, thus protecting the thermolabile volatile compounds from degradation, while the yield of oxygen-containing constituents (e.g., oxygenated monoterpenes) is increased and EO is recovered in an eco-friendly manner with reduced energy consumption [12,13,14]. Although microwave-assisted processes have already been widely used for the extraction of bioactive compounds (e.g., phenolics) from various materials, their use for EO distillation is quite limited. This may be attributed to the special equipment required to perform this distillation process (high operational costs) as well as themaintenance of the system. In addition, this method can be occasionally characterized by low selectivity, so optimization is required for the recovery of compounds of interest. The recovery of volatile compounds can be performed either from fresh or dry (rehydrated) plant matrices, employing solvent-free microwave extraction (SFME), microwave-assisted hydro-distillation (MAHD) and microwave hydro-diffusion and gravity (MHG) [7].

All the above distillation processes (HD, SD and MAD) result in two different types of products, the primary (EO) and the secondary (also referred as by-products), which include the remaining solid residue (distilled biomass), the resulting water that was in contact with the plant material during distillation (wastewater) and the water in the distillation separator where the EO was collected (hydrolat/hydrosol) [15]. The remaining solid biomass is composed of the distilled plant tissues, leaves, flowers, stalks, etc., and is considered as waste, with no important direct industrial value without further processing. Lately, the eco-friendly valorization of these by-products has been increased, since their disposal as waste causes environmental and economic problems, due to significant waste amounts generated by the processing industries [15]. In addition, these agricultural wastes can serve as an inexpensive and sustainable source of phenolic compounds that could be exploited as natural/safe antioxidants [16]. 

Medicinal and aromatic plants (MAPs) contain numerous compounds possessing antioxidant properties, like phenols, terpenes, nitrogen compounds, etc. During the EO extraction, most of these phenolic compounds remain in the solid plant residue, as they are non-volatile and non-degradable with thermal treatment. The selection of one of the aforementioned distillation methods might affect the chemical composition and biological activity of both EOs and their by-products.

Extensive studies have been carried out regarding the influence of different extraction methods on the yield and quality of EOs from MAPs [7,8,9,10,17,18,19]. Although several studies have reported the presence of bioactive compounds in solid residues remaining after EOs extraction from MAPs [20,21,22,23], limited information is available about the effect of the extraction methods on the bioactive composition of their solid waste. 

To address these issues, the aim of the present study was to investigate the effect of EO distillation methods on the profile of bioactive compounds and the antioxidant activity of the distilled solid residues of three common MAPs of the Mediterranean region: rosemary (*Rosmarinus officinalis* L.), Greek sage (*Salvia fruticosa* L.) and spearmint (*Mentha spicata* L.). The comparison was made in terms of the different distillation methods (conventional vs. green), namely hydro-distillation (HD), steam-distillation (SD) and microwave-assisted hydro-distillation (MAHD). As control groups, the initial raw plant materials were used in each case (prior to essential oil distillation).

## 2. Results and Discussion 

### 2.1. TPC, TFC and Antioxidant Activity of Distilled Solid By-Products

One of the main by-products of EO distillation is the solid residue, which remains after the recovery of EOs. As reported by previous studies, this solid residue is rich in valuable bioactive components such as phenolic compounds [15,16]. Results of the total phenolic content (TPC) and total flavonoid content (TFC) of the solid residue extracts are presented in Figure 1. Regardless of the plant material, the lowest TPC (23.66–25.84 mg GAE/g) and TFC (27.47–29.49 mg CATE/g) were observed in the extract obtained by HD (*p* ≤ 0.05), while the highest TPC (47.63–86.25 mg GAE/g) and TFC (68.78–145.93 mg CATE/g) were reported in the extracts by MAHD solid residues. The negative effect of HD in the TPC of solid residues could be attributed to the prolonged distillation time and the effect of the solvent (water), since several bioactive compounds could have been simultaneously extracted and remained in the respective wastewater, limiting the amount of these compounds in the solid residue [24,25]. For this reason, wastewater is considered a valuable distillation by-product and could be valorized for the recovery of important bioactive components. 

In addition, the TPC (41.93–68.44 mg GAE/g) and TFC (60.40–101.75 mg CATE/g) values of all three plant materials’ SD extracts were higher than the HD ones, but lower than the MAHD ones (*p* ≤ 0.05). It is worth mentioning that TPC (49.86–85.65 mg GAE/g) and TFC (66.28–123.75 mg CATE/g) of the raw plant material extracts (before distillation) were similar to the MAHD solid residue extracts. This could be due to the mild distillation conditions (temperature, time, pressure) during the process of MAHD [8]. In this way, the phenolic compounds of the plant materials remained almost stable and were not extracted to a great extent during MAHD compared to HD and SD, leading to a phenolic-rich distillation residue. 

As presented in Table 1, the antioxidant activities of the solid residue extracts, as determined by different assays (ABTS, DPPH and FRAP), were affected by the distillation method. A similar pattern was observed in the extracts obtained from HD solid residues, regardless of the plant material, which presented the lowest values in all assays (*p* ≤ 0.05). This may be attributed to the distillation conditions and, mainly, the contact of plant material with distillation water, probably leading to partial dissolution of hydrophilic phenolics. Additionally, prolonged distillation time and temperature during HD could be detrimental for the thermolabile bioactive compounds present in the plant materials, resulting in their degradation. Moreover, as mentioned above, many compounds could have remained in the respective wastewater, thus limiting the content of compounds available for extraction. 

In the ABTS assay, the antioxidant capacity of the extracts followed the pattern HD < SD < raw ≤ MAHD. It is worth noting that the hydrogen donating ability of the extracts against ABTS^·+^ was similar to the ones produced from raw plant materials and MAHD. This highlights the advantage of MAHD over conventional distillation methods, regarding the conditions applied, since limited distillation times offer protection to the bioactive compounds against degradation. Similarly, in the DPPH assay, the extracts’ antioxidant activity increased significantly in the following order: HD < SD < raw ≤ MAHD, apart from rosemary, where the respective SD solid residue extract presented the highest DPPH· scavenging activity (*p* ≤ 0.05). Unlike the other two, in the FRAP assay, no specific pattern was observed, and the antioxidant activity values varied corresponding to the plant species. Rosemary extracts from the SD solid residue presented the highest ability to donate electrons and reduce Fe^3+^ to Fe^2+^ (*p* ≤ 0.05), followed by MAHD, raw material and HD extracts. In the case of Greek sage, the FRAP values of MAHD and raw extracts were not statistically different, whereas the latter also did not differ from SD. However, the values between the MAHD and SD extracts were significantly different (*p* ≤ 0.05). The electron donation ability of spearmint extracts presented the following pattern: HD < SD < MAHD < raw (*p* ≤ 0.05). Other authors have also noted that MAE techniques are better for the extraction of phenolic compounds with antioxidant capacity, such as rosmarinic acid from plants of the Lamiaceae family like rosemary, oregano, peppermint and thyme [26,27,28]. The determination of the antioxidant capacities of extracts is important for their application in various food systems and highlights the protective role of these bioactive compounds against oxidation phenomena.

### 2.2. Polyphenolic Composition of Raw and Distilled Solid Residues

#### 2.2.1. Identification

The raw plant material (before distillation) and the solid residues obtained by HD, SD and MAHD for each species (rosemary, Greek sage and spearmint) showed similar phenolic profiles in their major compounds. However, significant differences were observed among the species (Table 2). A representative LC-MS chromatogram of phenolic compounds identified in rosemary, Greek sage and spearmint extracts from SD solid residues is shown in Figure 2. A total of 26 out of 29 major phenolic compounds were successfully identified in the phenolic extracts of solid residues, as listed in Table 2. 

In all extracts studied (rosemary, Greek sage and spearmint), the major phenolic compound identified was rosmarinic acid (caffeic acid dimer), showing a pseudo-molecular ion [M-H]- at *m*/*z* 359 (peak 12). Other major phenolic compounds in rosemary and Greek sage were carnosol and carnosic acid (phenolic triterpenes) with pseudo-molecular ion [M-H]- at *m*/*z* 329 (peak 26) and *m*/*z* 331 (peak 28), respectively, identified by comparing its UV spectra and retention times with those of the reference substances. Peaks 6, 8, 9, 11, 13, 15, 16, 17 and 19 presented a similar profile with maximum absorbance at ~280 nm and ~325 nm and were characteristic of caffeic acid trimers (salvianolic acids A, C, K, J and yunnaneic acid F) and caffeic acid tetramers (salvianolic acids B, E, I and L). This identification was based on the available literature data for these compounds, except for salvianolic acid B, for which a commercial standard was used. Specifically, three peaks (8, 13 and 19) presented the same molecular ion [M-H]—at *m*/*z* 717. Also, the MS spectra of these three compounds showed other fragment ions at *m*/*z* 537 (M-H-caffeic acid), *m*/*z* 519 (M-H-danshensu) and *m*/*z* 493 (M-H-caffeic acid-CO2), suggesting that these peaks could be assigned to salvianolic acid B (peak 13), salvianolic acid E (peak 19) and salvianolic acid L (peak 8), respectively [28,29,30]. Peak 15 presented an adduct ion at *m*/*z* 537 (M+HCOO), which, together with characteristic fragment ions at *m*/*z* 493 (M+HCOO-COO) and *m*/*z* 359, was identified as salvianolic acid C [26], while peak 16 presented a pseudo-molecular ion at *m*/*z* 493 that was identified as salvianolic acid A [29,30] and peak 17 presented a pseudo-molecular ion at *m*/*z* 537 as salvianolic acid J [30]. In addition, peak 6 showed a pseudo-molecular ion [M-H]- at *m*/*z* 597 that was identified as yunnaneic acid F, according to literature data [26], while peak 11 showed a pseudo-molecular ion [M-H]- at *m*/*z* 555 that was identified as salvianolic acid K [31]. Peak 9, with *m*/*z* 537 and the other fragments *m*/*z* 519 [M-H-H_2_O] and *m*/*z* 339 [M-H-H_2_O-caffeic acid], showed a UV max at 287 and 324 nm and was identified as salvianolic acid I; it presented only in salvia [31]. Most of the salvianolic acid isomers were identified in spearmint extracts. However, salvianolic acid I and K were identified only in Greek sage extract, and yunnaneic acid F only in rosemary extract. 

Peak 1 showed maximum absorbance at 320 nm and pseudo-molecular ion at *m*/*z* 191 and was identified as quinic acid, while peak 2 presented maximum absorbance at 280 nm and the pseudo-molecular ion at *m*/*z* 191 was characteristic of citric acid; both existed in all extracts studied. Peak 3 presented in rosemary and spearmint extracts was identified as danshensu, showing a pseudo-molecular ion [M-H]- at *m*/*z* 197, based on the literature data [25,29].

Peaks 4 and 7 presented a pseudo-molecular ion [M-H]- at *m*/*z* 305 and *m*/*z* 461, respectively, and were identified as gallocatechin isomer [28,32] and luteolin-7-*O*-glucuronide, respectively; both existed in all extracts studied. Peak 5 showed a pseudo-molecular ion [M-H]- at *m*/*z* 593 that was identified as luteolin-7-*O*-rutinoside, existing in Greek sage and spearmint extracts, while peak 10 presented a pseudo-molecular ion [M-H]- at *m*/*z* 609 that was identified as hesperidin. The extracts of rosemary and Greek sage were characterized by the presence of rosmanol isomers (peaks 22, 23, 24, 27 and 29), yielding *m*/*z* at 345 [25,31,32]. Moreover, rosmanol was also identified in spearmint extracts at peak 29 [29].

#### 2.2.2. Quantification

Most of the phenolic compounds identified by LC/ESI-MS were also quantified, and the results of the major and minor phenolic compounds are shown in Table 3. The total amount of the phenolic compounds quantified by LC-MS in rosemary, Greek sage and spearmint extracts (raw material and solid residues from three different distillation methods) ranged from 69.31 to 170.61 mg/g, 73.70 to 116.78 mg/g and 69.25 to 161.63 mg/g, respectively. The highest content of total phenolics quantified by LC-MS in raw materials was reported in spearmint, followed by rosemary and Greek sage. The amount of total phenolics in solid residues from HD, for all plants, was the lowest among the three distillation methods applied. SD solid residue extracts from rosemary and Greek sage extracts retained the highest total phenolics, while the spearmint extract obtained from MAHD yielded the highest phenolic content. Rosmarinic acid, carnosol and carnosic acid were the major phenolic compounds in rosemary and Greek sage extracts, constituting more than 95% of the total quantified phenolic compounds (Figure 3). However, rosmarinic acid and salvianolic acids isomers were the main phenolic compounds in spearmint extract, constituting more than 90% of the total phenolic compounds. 

Raw plant materials had higher rosmarinic acid content compared to solid residues from HD and SD, whereas solid residues from MAHD presented the highest rosmarinic acid content among all treatments. Increased concentration of rosmarinic acid in MAHD solid residue extracts was also observed by Navarrete et al. [33], and it was attributed to the effects of microwaves that may raise the temperature in plant tissue, leading to liberation of phenolic compounds to the solvent. A previous study on *Origanum vulgare* L. spp. *hirtum* solid residue extracts obtained by MAHD and HD showed that the highest rosmarinic acid content was found in the MAHD extract at a power level of 600 W [34]. In our study, the initial (raw material) rosmarinic acid content of 40.70 mg/g in rosemary decreased significantly to 14.00 mg/g and 32.33 mg/g after HD and SD, respectively, followed by a significant increase to 51.34 mg/g after. Similar results were also observed by Wollinger et al. [35], who studied the effect of HD and SD on the antioxidant compound content of rosemary distillation residues. In the case of the present distillation experiments, it was similarly found that the initial (raw material) rosmarinic acid content of 36.09 mg/g in spearmint decreased significantly to 24.24 mg/g and 12.75 mg/g after SD and HD, respectively, followed by a small increase to 44.60 mg/g after MAHD. In the case of Greek sage, the initial rosmarinic acid content of 52.46 mg/g in the raw material decreased significantly to 26.05 mg/g and 49.92 mg/g after HD and SD, respectively; however, a significant increase was observed after MAHD.

In rosemary extracts, carnosol content after SD (45.05 mg/g) was double the amount obtained after HD and MAHD. A similar pattern was observed for carnosic acid content, which was around two times higher after SD (83.47 mg/g) compared to HD and MAHD. Our findings are in contrast to the results of Navarrete et al. [33], who indicated higher extraction yields and rates from rosemary leaves after solvent-free microwave extraction. However, our findings are in agreement with those of Jacotet-Navarro et al. [36], who noticed a decrease of carnosic acid for all microwave-assisted extraction techniques. The initial carnosol content of 37.44 mg/g in the raw material decreased significantly to 22.67 mg/g and 20.12 mg/g after HD and MAHD, respectively; however, a significant increase to 45.05 mg/g was observed after SD. On the other hand, the initial carnosic acid content of 48.12 mg/g in the raw material increased significantly to 90.67 mg/g and 57.36 mg/g after SD and HD, respectively, but a small decrease to 46.81 mg/g was reported after MAHD. In Greek sage extracts, the distillation method applied had no effect on carnosol content; however, a substantial increase was observed in carnosic acid. Specifically, the initial carnosic acid content of 8.56 mg/g in the raw material increased significantly to 26.04 mg/g, 32.65 mg/g and 40.83 mg/g after HD, MAHD and SD, respectively. 

In spearmint extracts, the initial total salvianolic acids content (sum of all isomers) of 102.93 mg/g in the raw material decreased significantly to 53.93 mg/g and 64.47 mg/g after HD and SD, respectively, followed by an insignificant increase to 107.65 mg/g after MAHD. Thus, the heating treatments in spearmint had a negative effect on salvianolic acids content, which was in agreement with other studies indicating the degradation of salvianolic acid with the increase of temperature above 30 °C for more than 30 min [37]. However, in our study, we found that MAHD had a positive influence on the recovery of salvianolic acids. Similarly, Putnik et al. [38] suggested that elevated temperatures (60 and 80 °C vs. 30 and 50 °C) yielded significantly higher concentrations of salvianolic acid K after microwave-assisted extraction of sage leaves. Concerning the distribution of salvianolic acid isomers, salvianolic acid E (8.35–54.74 mg/g) was the major component existing in spearmint extracts, followed by salvianolic acids A (9.50–21.04 mg/g) and L (8.25–22.32 mg/g), whereas minor components were salvianolic acids B (5.08–8.85 mg/g) and C (5.20–6.62 mg/g). 

Minor phenolic compounds quantified in rosemary, Greek sage and spearmint extracts were quinic acid (0.20–6.42 mg/g), caffeic acid (0.12–0.39 mg/g), vicenin-2 (0.08–0.32 mg/g), luteolin-7-*O*-glucuronide (0.11–2.25 mg/g) and luteolin-7-*O*-glucuronide (trace–3.66 mg/g). In rosemary and spearmint extracts, hesperidin was also quantified, and its content ranged from 0.18 to 1.83 mg/g. The total content of minor phenolic compounds in rosemary, Greek sage and spearmint extracts ranged from 1.62 to 7.22 mg/g, 0.66 to 2.63 mg/g and 2.57 to 13.32 mg/g, respectively. The lowest value for all treatments appeared in solid residues deriving from HD, whereas the highest value was in SD extracts.

### 2.3. Pearson Correlation Analysis

In order to better evaluate the relationships between the antioxidant activities, TPC, TFC and the main phenolic compounds, Pearson correlation assays were performed (Table 4). The TPC was highly and positively correlated with TFC (r = 0.867, *p* ≤ 0.001) and salvianolic acids (r = 0.867, *p* ≤ 0.001), while low and positive correlation was observed with rosmarinic acid (r = 0.414, *p* ≤ 0.05) and carnosol (r = 0.420, *p* ≤ 0.05) and no correlation was observed with carnosic acid. In addition, TPC was strongly and positively correlated with antioxidant activity by DPPH (r = 0.986, *p* ≤ 0.001) and FRAP (r = 0.982, *p* ≤ 0.001) tests, whereas low correlation appeared with the ABTS test (r = 0.418, *p* ≤ 0.05). Similar to TPC, high correlations were observed between TFC-DPPH (r = 0.986, *p* ≤ 0.001), TFC-FRAP (r = 0.967, *p* ≤ 0.001) and TFC-salvianolic acids (r = 0.879, *p* ≤ 0.001), whereas low correlation appeared between TFC-ABTS (r = 0.366, *p* ≤ 0.05), TFC-RMA (r = 0.405, *p* ≤ 0.05) and TFC-carnosol (r = 0.416, *p* ≤ 0.05).

Concerning rosmarinic acid, it was highly and positively correlated with ABTS (r = 0.797, *p* ≤ 0.001), while low correlation was found with DPPH (r = 0.345, *p* ≤ 0.05) and FRAP (r = 0.389, *p* ≤ 0.05). This may be attributed to the synergistic effects of other bioactive compounds present in the plant, which may contribute to the overall observed antioxidant capacity [39]. In this study, carnosol was medium and positively correlated with DPPH (r = 0.476; *p* ≤ 0.05) and FRAP (r = 0.413; *p* ≤ 0.05), but no correlation was noticed with ABTS. On the other hand, carnosic acid was positively correlated to carnosol (r = 0.827; *p* ≤ 0.001), while no correlation was noted among antioxidant assays. Thus, it could be concluded that there are other compounds different from phenolic diterpenoids that contribute to the antioxidant activity. This may be attributed to the drawbacks of the assays and the chemical nature of the active substances contained in the extracts. No correlation between the two phenolic diterpenes with DPPH and FRAP assays has been previously reported [40].

### 2.4. Multivariate Data Analysis

Principal components analysis (PCA) was performed using a data matrix composed of 108 data points (9 variables × 12 observations) to display the grouping among distilled solid residues based on their phenolic components (TPC, TFC, rosmarinic acid, sum of salvianolic acids isomers, carnosol and carnosic acid) and three antioxidant attributes (DPPH, ABTS and FRAP values) (Figure 4). The first two principal components (PC1 and PC2) explained 57.7% and 31.0% of the total variability, respectively, representing 88.7% of the total variance. 

According to the score values of PCA, the solid residues obtained by HD, SD and MAHD, as well the raw plant material (before distillation), were classified into four different groups (Figure 4A). In particular, distilled solid residues obtained after HD could be well distinguished from those obtained by SD, MAHD and raw material (before distillation). They were allocated in the lower left quadrant, presenting a rather similar response with lower values at most evaluated parameters. However, the majority of the other samples were allocated on the upper right quadrant. In addition, bioactive profile of the solid residues obtained by MAHD and SD showed non-noticeable differences with raw plant material.

Moreover, all the spearmint extracts could be well distinguished from rosemary and Greek sage extracts and were positioned on the left of the score values in Figure 4B. As regards the effect of the distillation methods on the bioactive profile of the solid residue of spearmint, the raw and MAHD-treated samples were positioned close to each other in Figure 4B, indicating that they presented some similarities. On the other hand, all the rosemary and Greek sage extracts were close to each other; therefore, they were classified into the same group. 

In addition, hierarchical clustering visualized in the heatmap (Figure 5) revealed the grouping of raw and solid residues, according to their phenolic profile, into two main clusters: the first cluster comprised all spearmint extracts (raw and treated), while the second one included two subgroups containing the extracts of rosemary and Greek sage. More specifically, the first subgroup contained the rosemary extracts excluding that of MAHD, exhibiting high content of carnosol and carnosic acid, whereas the second subgroup contained the Greek sage extracts except that of HD. 

## 3. Materials and Methods

### 3.1. Plant Materials

In the current study, the extraction processes SD, HD and MAHD were applied on three known EO plants of the Lamiaceae family: *Rosmarinus officinalis* L. (rosemary), *Salvia fruticosa* L. (Greek sage) and *Mentha spicata* L (spearmint). The aerial parts of the three species were collected between April and May 2022 from the field MAPs’ collection of the Institute of Plant Breeding and Genetic Resources–ELGO-DIMITRA (Thermi, Thessaloniki, Greece). Immediately after harvesting, the fresh plant material was comminuted and subjected to distillation for the recovery of EOs. The average moisture content of the fresh plant material was measured at 68.2% for rosemary, 79.7% for Greek sage and 73.5% for spearmint, respectively.

### 3.2. Distillation of Aromatic Plants

#### 3.2.1. Steam-Distillation (SD)

Fresh plant material (2000 g) of the three plant species was placed in a 20 L-capacity stainless steel distillation unit (Albrigi, Mod. Plus) and was connected to a boiler. The steam was produced in the boiler by electrical heating of the distilled water. The vapor from the boiler went through the fresh plant material. The operation lasted approximately 2 h and after the recovery of the EOs, the remaining solid residue was collected.

#### 3.2.2. Hydro-Distillation (HD)

Fresh material (50 g) of each plant species was placed in a flask containing 500 mL of distilled water and was connected to a Clevenger distillation apparatus (Heating isomantel, 700 W). The duration of the distillation process was approximately 2 h, and, after the recovery of the EOs, the remaining solid residue was collected. The HD process was repeated for each plant under the same conditions.

#### 3.2.3. Microwave-Assisted Hydro-Distillation (MAHD)

The MAHD was performed in a laboratory microwave ETHOS X oven (Milestone, Sorisole, Italy) designed for fragrances extraction. The apparatus is composed of a microwave reactor of 2.45 GHz, equipped with two magnetrons with a maximum delivered power of 1800 W (2 × 950 W) and an infrared sensor monitoring the temperature. All the parameters regarding the procedure, such as temperature, time, power, and pressure were controlled by the respective software. The distillation was performed using 500 g of fresh plant material (previously hydrated with 0.5 L of water for 30 min) that was placed in a glass reactor vessel (Pyrex) of 2 L capacity and then closed with a glass cover. The experiment was conducted at atmospheric pressure under a heating–cooling cycle, including heating at 1000 W for 5 min, followed by 500 W for 30 min, and finally cooling for 10 min. A controlled water-cooling process outside the microwave oven was used for the distillate’s condensation. The experiments were performed in duplicate for each plant under the same conditions. After the recovery of the EO, the remaining solid residue was collected.

### 3.3. Pretreatment of Resultant Distilled Solid By-Products

The wet solid residues remaining after the EO extraction were sun-dried for 24 h, followed by oven-drying at 40 °C until their moisture contents reached approximately 10%, as evaluated by a moisture analyzer (Ohaus, MB90, Parsippany, NJ, USA). Then, the dried plant samples were ground to pass through a 0.5 mm sieve in a laboratory mill (Retsch, Model ZM1000, Haan, Germany) and stored at 4 °C until further analysis.

### 3.4. Extraction of Phenolics from Distilled Solid Residues 

A total of 0.05 g dried and ground solid residue samples were subjected to extraction using 10 mL of 70% methanol with the aid of an ultrasonic bath (frequency 37 kHz, model FB 15051, Thermo Fisher Scientific Inc., Loughborough, England) for 15 min at room temperature. The extracts were then centrifuged at 10,000× *g* for 10 min at 4 °C, the supernatant was collected, and the extraction was repeated one more time. The clear supernatants were mixed, filtered through PTFE syringe filters with a porosity of 0.22 µm and stored at −20 °C until analysis.

### 3.5. Determination of Total Phenolic (TPC) and Flavonoid Content (TFC) 

The TPC of the above phenolic extracts was determined by the Folin-Ciocalteu spectrophotometric procedure with a slight modification [41]. Briefly, 200 μL of extract was added to 800 μL of 1:10 diluted Folin–Ciocalteu reagent and, after 2 min, 2 mL of sodium carbonate (75 g/L) was added, and the final volume was adjusted to 10 mL with distilled water. The absorbance was measured at 725 nm after 1 h of incubation in the dark at room temperature, and the results were expressed as milligrams of gallic acid equivalents per g of plant dry weight (mg GAE/g).

TFC was determined using the colorimetric assay with aluminum chloride, as described by Bao et al. [42]. Briefly, 300 μL of phenolic extract was mixed with 225 μL of sodium nitrite (50 g/L), followed by the addition of 225 μL of 10% aluminum chloride hexahydrate (100 g/L) and 750 μL of NaOH (2 N). After 20 min of incubation, the absorption was recorded at 765 nm. TFC and the results were expressed as milligrams of catechin equivalents per g of plant dry weight (mg CATE/g).

### 3.6. Determination of Antioxidant Activity of Phenolic Extracts

#### 3.6.1. ABTS Radical Scavenging Assay

The radical scavenging activity of phenolic extracts against ABTS radical cations was determined according to Re et al. [43]. Briefly, 100 μL of phenolic extract was mixed with 3.9 mL of the ABTS^+^ solution, and after 4 min, the absorbance was recorded at 734 nm against a control. The ABTS results were expressed as mg Trolox equivalents (TE) per g of plant dry weight (mg TE/g).

#### 3.6.2. DPPH Radical Scavenging Assay

The DPPH (2,2-diphenyl-1-picrylhydrazyl) scavenging activity of phenolic extracts was measured according to a previous report [44], with slight modifications. Briefly, 100 μL of phenolic extract was mixed with 2.85 mL of freshly prepared 0.1 mM DPPH in methanol, and the decrease in absorbance was measured at 516 nm after 5 min of reaction. The DPPH values were expressed as mg TE/g of plant dry weight.

#### 3.6.3. Ferric Reducing Antioxidant Power (FRAP) Assay

FRAP activity of the phenolic extracts was evaluated based on the method of Benzie and Strain [45]; 100 μL of phenolic extract was mixed with 3 mL of FRAP solution at 37 °C. After 4 min exactly, the absorbance at 593 nm was reordered against a control, and the FRAP values were expressed as mg TE/g of plant dry weight. 

### 3.7. HPLC-DAD-MS Analysis of Phenolics from Raw and Distillated Solid Residues

The chromatographic analysis was carried out on a Shimadzu Nexera HPLC system (Kyoto, Japan) consisting of LC-30AD pumps, a DGU-20A5 degasser, a CTO-20AC column oven, a SIL-30AC auto injector, a SPD-M40 diode array detector (DAD) and a single quadrupole mass spectrometer (LCMS-2020), which was operated with an electrospray ionization (ESI) interface. Separations were made with a Poroshell 120 EC-C_18_ analytical column (4.6 × 150 mm, 4 µm) according to a method described in the literature [20]. The mobile phases were acidified water (0.1% formic acid, *v*/*v*) and acetonitrile as eluents A and B, respectively. The chromatographic method consisted of the following multistep linear gradient: 0 min, 15% B; 5 min, 25% B; 10 min, 35% B; 28 min, 60% B; 35.00 min, 100% B; 35.01 min, 15% B; 42 min, 15% B. The injection volume was 10 μL and the column temperature was maintained at 35 °C. The flow rate was set at 0.5 mL/min throughout the gradient.

Mass detection was carried out in a negative electrospray ionization (ESI) mode. The optimum values of the ESI-MS parameters were interface voltage, +4.5 kV; curved desolvation line (CDL) voltage, 20 V; block heater temperature, 200 °C; CDL temperature, 250 °C; nebulizing gas (nitrogen) flow rate, 1.5 L/min; drying gas flow, 15 L/min; nebulizing gas pressure, 11 psi. Mass acquisitions were performed by either full scan mode in the range of 100–1000 *m*/*z* or targeted selective ion monitoring mode (SIM). Data acquisition and processing was carried out using Lab Solutions LC-MS software (Shimadzu, Kyoto, Japan).

For identification, the total ion current (TIC) profile was produced by monitoring the intensity of all the ions produced and acquired in every scan during the chromatographic run. The main phenolic compounds of the samples were identified by comparing their retention time, UV profile and the mass spectra of unknown peaks with those of authentic standards or literature data. For quantitative measurements, a TIC profile was produced in SIM mode, using the calibration curves of corresponding standard solutions. In the case of salvianolic acids isomers, quantification was based on the calibration curve of salvianolic acid B, due to unavailability of standards.

### 3.8. Statistical Analysis

The values were expressed as mean ± standard deviations of three independent measurements. Data were tested using SPSS Statistics software version 19 (IBM SPSS Inc., Chicago, IL, USA). One-way analysis of variance (ANOVA) was used to test for differences among the means for different extracts, according to Duncan’s post-hoc test. Linear Pearson’s coefficients were evaluated to determine the correlation between bioactive compounds and antioxidant activity of the extracts among different treatments; differences at *p* ≤ 0.05 were considered significant. Principal component analysis (PCA) and the heatmap were evaluated using the web tool ClustVis (https://biit.cs.ut.ee/clustvis, accessed on 21 November 2022) for visualizing the clustering of multivariate data. 

## 4. Conclusions

In the present study, MAHD was investigated as a green approach for the recovery of EOs from three aromatic plants and was compared to conventional distillation methods. Based on the results, MAHD was an effective distillation method in terms of distilled solid residues, as it presented a much higher content of bioactive compounds than HD and SD methods. In fact, HD solid residue extracts had the lowest TPC and TFC values, followed by SD, whereas the respective values for MAHD solid residue extracts were higher and, in some cases, similar to that of the raw plant materials (before distillation). Major phenolic compounds were successfully identified through LC-MS in the phenolic solid residue extracts of rosemary, Greek sage and spearmint. Rosmarinic acid, carnosol and carnosic acid were the main compounds identified and quantified in rosemary and Greek sage extracts, while in spearmint extracts, the main identified compounds were rosmarinic acid and salvianolic acids isomers. 

Solid distillation residues are rich in bioactive compounds, mainly polyphenols, that can be further exploited for applications in several industrial sectors, such as food, cosmetics and pharmaceuticals. Their valorization through green extraction and distillation methods promotes sustainable development and provides new alternative sources for the recovery of important bioactive components with antioxidant activity. 

## Figures and Tables

**Figure 1 molecules-27-09058-f001:**
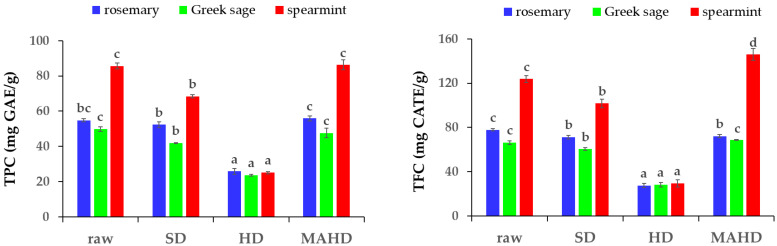
Total phenolic content (TPC) and total flavonoid content (TFC) in methanolic extracts from raw and distillation solid residues of rosemary, Greek sage and spearmint, remained after the recovery of essential oil by steam distillation (SD), hydro-distillation (HD) and microwave-assisted hydro-distillation (MAHD). Different letters among columns with the same color indicate statistically significant differences based on Duncan’s post-hoc test (*p* ≤ 0.05).

**Figure 2 molecules-27-09058-f002:**
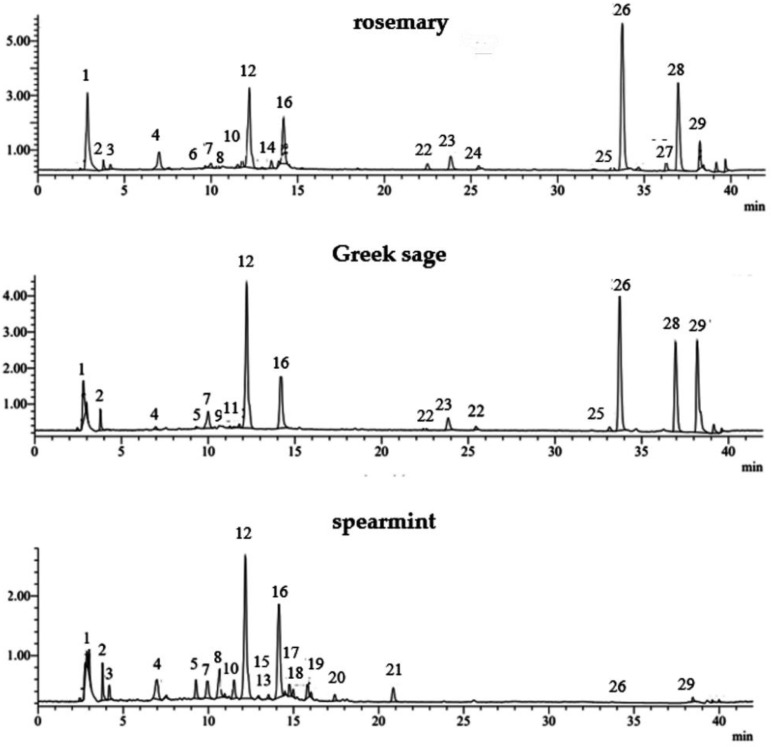
Mass chromatograms recorded in the negative ion mode for methanolic extracts from distillation residues of rosemary, Greek sage and spearmint, after the recovery of essential oil by steam distillation (SD). (1: quinic acid; 2: citric acid; 3: danshensu; 4: gallocatechin isomer; 5: luteolin-7-*O*-rutinosid; 6: yunnaneic acid F; 7: luteolin-7-*O*-glucuronide; 8: salvianolic acid L; 9: salvianolic acid I; 10: hesperidin; 11: salvianolic acid K; 12: rosmarinic acid; 13: salvianolic acid B; 14: caffeoyl-hexosyl-hexose; 15: salvianolic acid C; 16: salvianolic acid A (internal standard); 17: salvianolic acid J; 19: salvianolic acid E; 21: rosmarinic acid derivative; 22–24, 27 and 29: rosmanol or isomer; 26: carnosol; 28: carnosic acid; 18, 20 and 25: unknown).

**Figure 3 molecules-27-09058-f003:**
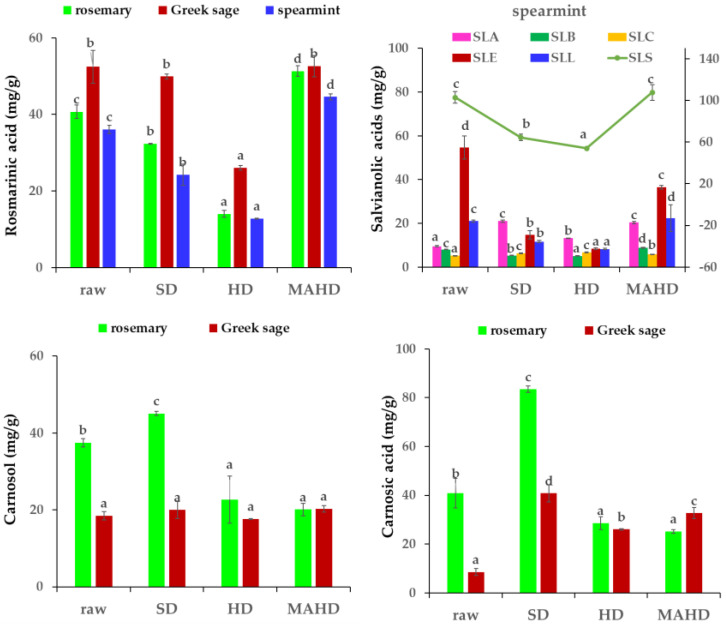
Major phenolic compound quantification in rosemary, Greek sage and spearmint distillation residues after the recovery of essential oils by steam-distillation (SD), hydro-distillation (HD) and microwave-assisted hydro-distillation (MAHD) by HPLC-DAD-MS. Abbreviations: SLA, salvianolic acid A; SLB, salvianolic acid B; SLC, salvianolic acid C; SLE, salvianolic acid E; SLL, salvianolic acid L; SLS, total salvianolic acids. Different letters among columns with the same color indicate statistically significant differences based on Duncan’s post-hoc test (*p* ≤ 0.05).

**Figure 4 molecules-27-09058-f004:**
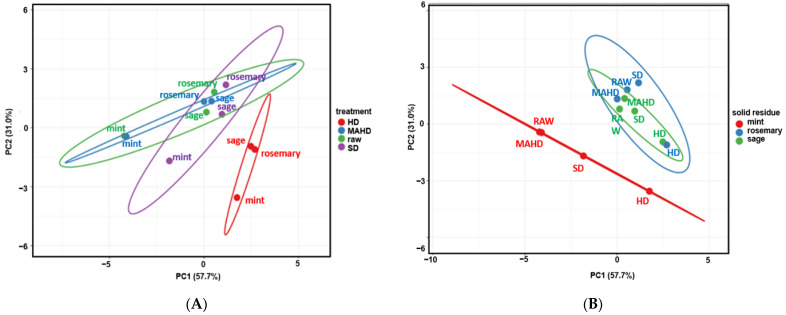
PCA score plots of rosemary, Greek sage and spearmint solid residues after the extraction of their essential oils by steam-distillation (SD), hydro-distillation (HD) and microwave-assisted hydro-distillation (MAHD). (**A**) different treatments; (**B**) different solid residues.

**Figure 5 molecules-27-09058-f005:**
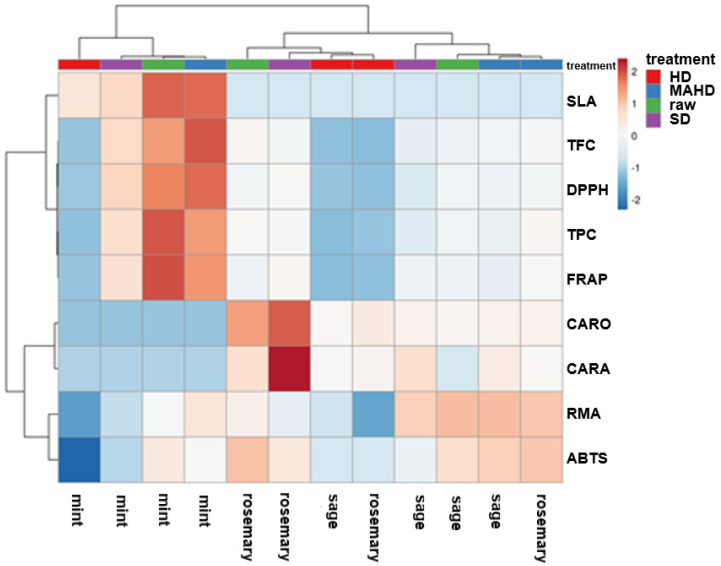
Heatmap explaining the different contents of rosmarinic acid (RMA), salvianolic acids (SLA), carnosol (CARO) and carnosic acid (CARA), as well as total phenolic content (TPC), total flavonoid content (TFC) and antioxidant activity as estimated by DPPH, ABTS and FRAP assays of rosemary, Greek sage and spearmint extracts of solid residues remaining after the extraction of their essential oils by steam-distillation (SD), hydro-distillation (HD) and microwave-assisted hydro-distillation (MAHD).

**Table 1 molecules-27-09058-t001:** In vitro antioxidant activity of methanolic extracts of rosemary, Greek sage and spearmint distillation solid residues after the recovery of essential oils by steam-distillation (SD), hydro-distillation (HD) and microwave-assisted hydro-distillation (MAHD).

Solid Residue	Distillation Method	Antioxidant Activity (mg TE/g)		
		ABTS	DPPH	FRAP
*rosemary*	raw	216.15 ± 7.77 ^C^	91.17 ± 1.49 ^B^	88.50 ± 0.83 ^B^
	SD	188.90 ± 1.29 ^B^	96.13 ± 3.54 ^C^	102.61 ± 1.53 ^D^
	HD	139.50 ± 4.52 ^A^	47.87 ± 0.95 ^A^	32.21 ± 0.70 ^A^
	MAHD	214.16 ± 3.62 ^C^	89.79 ± 1.55 ^B^	97.76 ± 1.32 ^C^
*Greek sage*	raw	200.75 ± 3.66 ^C^	88.05 ± 3.89 ^C^	86.20 ± 0.00 ^BC^
	SD	158.44 ± 4.21 ^B^	72.19 ± 1.16 ^B^	88.07 ± 1.17 ^C^
	HD	139.88 ± 2.05 ^A^	49.61 ± 0.52 ^A^	30.07 ± 1.84 ^A^
	MAHD	210.10 ± 0.72 ^D^	86.94 ± 1.04 ^C^	83.39 ± 2.91 ^B^
*spearmint*	raw	182.03 ± 5.44 ^D^	151.38 ± 2.30 ^C^	194.11 ± 7.88 ^D^
	SD	128.29 ± 2.83 ^B^	125.18 ± 3.85 ^B^	130.20 ± 5.00 ^B^
	HD	65.64 ± 8.08 ^A^	50.90 ± 4.61 ^A^	33.90 ± 7.07 ^A^
	MAHD	169.71 ± 6.44 ^C^	161.65 ± 5.96 ^D^	174.67 ± 5.06 ^C^

Different superscript letters in each column (for each solid residue) indicate statistically significant differences based on Duncan’s post hoc test (*p* ≤ 0.05); TE: Trolox equivalents.

**Table 2 molecules-27-09058-t002:** List of tentative major compounds identified by LC-DAD-MS in rosemary (R), Greek sage (S) and spearmint (M) distillation residues after the recovery of essential oils through steam-distillation (SD), hydro-distillation (HD) and microwave-assisted hydro-distillation (MAHD).

Peak	RT (min)	UV λmax (nm)	[M-H]-(*m*/*z*)	Other Fragments	Tentative Identification		Solid Residues			
						Ref.	Raw	SD	MAHD	HD
1	2.9	320	191	133	quinic acid	st	R, S, M	R, S, M	R, S, M	R, S, M
2	3.8	280	191	147	citric acid	st	R, S, M	R, S, M	R, S, M	R, S, M
3	4.2	280	197	395, 179	danshensu	[25,29]	R, Μ	R, M	R, M	R, M
4	7.0	284, 312	305	387, 264, 198	gallocatechin isomer	[28,32]	R, S, M	R, S, M	R, S, M	R, S, M
5	9.3	260, 345	593	430, 259, 166	luteolin-7-*O*-rutinoside	st	S, M	S, M	S, M	S, M
6	9.6	280, 325	597	435, 329	yunnaneic acid F	[25]	R	R	R	R
7	10.0	260, 345	461	923, 479, 285	luteolin-7-*O*-glucuronide	st	R, S, M	R, S, M	R, S, M	R, S, M
8	10.6	283, 340	717	579, 359	salvianolic acid L	[26,29]	R, M	R, M	R, M	R, M
9	11.2	287, 341	537	519, 339	salvianolic acid I	[31]	S	S	S	S
10	11.5	283	609	301	hesperidin	st	R, M	R, M	R, M	R, M
11	11.6	286, 325	555	493, 445, 359	salvianolic acid K	[31]	S	S	S	S
12	12.2	329, 285 sh	359	719, 405, 161	rosmarinic acid	st	R, S, M	R, S, M	R, S, M	R, S, M
13	12.8	287, 325	717	537, 519, 493, 359	salvianolic acid B	st	M	M	M	M
14	13.4	266, 340	503	285	caffeoyl-hexosyl-hexose	[25]	R	R	R	R
15	13.5	325	491	537, 493, 359, 161	salvianolic acid C	[26]	M	M	M	M
16	14.2	266, 327 (240)	493 (137)	537, 359, 183	salvianolic acid A (IS)	[29,30]	S, M	S, M	S, M	S, M
17	14.5	283, 340	537	715, 441	salvianolic acid J	[29]	M	M	M	M
18	14.8	287, 325	593	413, 285	unknown	-	M	M	M	M
19	15.8	286, 325	717	537, 519, 493, 339	salvianolic acid E	[28,29]	M	M	M	M
20	17.4	282, 333	329	-	unknown	-	M	M	M	M
21	20.9	280, 345	359	329	rosmarinic acid derivative	[29]	M	M	M	M
22	22.5	286	345	301, 283	rosmanol or isomer	[25,31]	R, S	R, S	R, S	R, S
23	23.8	286	345	301, 283	rosmanol or isomer	[25,31]	R, S	R, S	R, S	R, S
24	25.4	286	345	301	rosmanol or isomer	[25,31]	R, S	R, S	R, S	R, S
25	33.2	-	359	329, 283	unknown	-	R, S	S	S	S
26	33.7	280	329	285	carnosol	st	R, S, M	R, S	R, S	R, S
27	36.3	279	345	301	rosmanol or isomer	[25,32]	R	R	R	R
28	37.0	280	331	287	carnosic acid	st	R, S	R, S	R, S	R, S
29	38.2	278	345	301	rosmanol or isomer	[25,29]	R, S, M	R, S	R, S	R, S

IS, internal standard (salicylic acid).

**Table 3 molecules-27-09058-t003:** Minor, major and total phenolic compounds (in mg/g) quantified in rosemary, Greek sage and spearmint distillation residues after the recovery of essential oils by steam-distillation (SD), hydro-distillation (HD) and microwave-assisted hydro-distillation (MAHD) by HPLC-DAD-MS ^1^.

Plant Material	Treatment	Phenolic Compounds								
		Minor							Major	Total
		Quinic Acid	Caffeic Acid	Vicenin-2	Hesperidin	Luteolin-7-*O*-glucuronide	Luteolin-7-*O*-rutinoside	Total	RMA + SLA + CARO + CARA	
**rosemary**	**raw**	4.90 ± 0.14 ^B^	0.15 ± 0.00 ^B^	0.11 ± 0.00 ^B^	0.58 ± 0.02 ^D^	0.23 ± 0.00 ^B^	<LOQ ^2^	5.98 ± 0.15 ^B^	122.00 ± 8.78 ^C^	127.99 ± 8.88 ^C^
	**SD**	6.42 ± 0.07 ^C^	0.24 ± 0.01 ^D^	0.11 ± 0.00 ^B^	0.18 ± 0.01 ^A^	0.27 ± 0.01 ^C^	<LOQ	7.22 ± 0.08 ^C^	163.40 ± 1.69 ^D^	170.61 ± 1.65 ^D^
	**HD**	1.15 ± 0.01 ^A^	0.12 ± 0.00 ^A^	0.08 ± 0.00 ^A^	0.24 ± 0.02 ^B^	0.11 ± 0.01 ^A^	<LOQ	1.62 ± 0.03 ^A^	67.69 ± 4.72 ^A^	69.31 ± 4.70 ^A^
	**MAHD**	4.78 ± 0.59 ^B^	0.17 ± 0.01 ^C^	0.11 ± 0.00 ^B^	0.34 ± 0.02 ^C^	0.23 ± 0.01 ^B^	<LOQ	5.63 ± 0.59 ^B^	99.31 ± 3.20 ^B^	104.94 ± 3.07 ^B^
**Greek sage**	**raw**	1.32 ± 0.26 ^B^	0.19 ± 0.03 ^B^	0.29 ± 0.02 ^B^	ND ^3^	0.38 ± 0.03 ^B^	0.30 ± 0.04 ^BC^	2.48 ± 0.31 ^B^	82.64 ± 4.81 ^B^	85.12 ± 4.65 ^B^
	**SD**	1.41 ± 0.01 ^B^	0.23 ± 0.00 ^C^	0.28 ± 0.01 ^B^	ND	0.46 ± 0.02 ^C^	0.25 ± 0.01 ^B^	2.63 ± 0.03 ^B^	114.16 ± 5.16 ^A^	116.78 ± 5.16 ^A^
	**HD**	0.20 ± 0.01 ^A^	0.13 ± 0.00 ^A^	0.14 ± 0.00 ^A^	ND	0.19 ± 0.00 ^A^	<LOQ	0.66 ± 0.01 ^A^	73.04 ± 0.83 ^B^	73.70 ± 0.83 ^C^
	**MAHD**	1.24 ± 0.03 ^B^	0.24 ± 0.00 ^C^	0.33 ± 0.02 ^C^	ND	0.50 ± 0.07 ^C^	0.32 ± 0.05 ^C^	2.62 ± 0.17 ^B^	108.92 ± 4.52 ^A^	111.54 ± 4.46 ^A^
**spearmint**	**raw**	4.91 ± 0.22 ^C^	0.26 ± 0.00 ^C^	0.18 ± 0.00 ^C^	1.83 ± 0.06 ^D^	2.25 ± 0.01 ^D^	3.66 ± 0.04 ^D^	12.58 ± 0.94 ^D^	139.01 ± 6.60 ^C^	152.34 ± 6.76 ^C^
	**SD**	4.02 ± 0.30 ^B^	0.39 ± 0.01 ^D^	0.15 ± 0.01 ^B^	0.89 ± 0.06 ^B^	2.04 ± 0.09 ^C^	1.84 ± 0.12 ^B^	11.92 ± 0.33 ^C^	88.71 ± 5.63 ^B^	100.63 ± 5.93 ^B^
	**HD**	0.95 ± 0.00 ^A^	0.17 ± 0.00 ^A^	0.11 ± 0.01 ^A^	0.42 ± 0.04 ^A^	0.76 ± 0.02 ^A^	0.83 ± 0.02 ^A^	2.70 ± 0.39 ^A^	66.88 ± 0.66 ^A^	69.25 ± 0.80 ^A^
	**MAHD**	4.17 ± 0.06 ^B^	0.22 ± 0.01 ^B^	0.19 ± 0.00 ^C^	1.17 ± 0.06 ^C^	1.89 ± 0.01 ^B^	2.64 ± 0.07 ^C^	7.10 ± 0.29 ^B^	152.57 ± 7.59 ^CD^	161.63 ± 4.05 ^CD^

^1^ Results are expressed as means values (mg/g) ± standard deviation; ^2^ limit of quantification; ^3^ not detected; different superscript letters in each column (for each plant material) indicate statistically significant differences based on Duncan’s post hoc test (*p* ≤ 0.05). RMA, rosmarinic acid; SLA, salvianolic acids; CARO, carnosol; CARA, carnosic acid.

**Table 4 molecules-27-09058-t004:** Pearson’s correlation coefficients (r) analysis between different assays (TPC, TFC, ABTS, DPPH and FRAP) and main phenolic compounds (RMA, CARO, CARA and SLA).

Parameters	TPC	TFC	ABTS	DPPH	FRAP	RMA	CARO	CARA	SLA
TPC	1	0.867 ***	0.418 *	0.985 ***	0.982 ***	0.414 *	0.420 *	ns	0.867 ***
TFC		1	0.366 *	0.986 ***	0.967 ***	0.405 *	0.416 *	ns	0.879 ***
ABTS			1	0.330 *	0.369 *	0.797 ***	ns	ns	0.913 ***
DPPH				1	0.976 ***	0.345 *	0.476 *	ns	0.861 ***
FRAP					1	0.389 *	0.413 *	ns	0.885 ***
RMA						1	ns	ns	0.954 ***
CARO							1	0.827 ***	ns
CARA								1	ns
SLA									1

ns, non-significant; * *p* ≤ 0.05; *** *p* ≤ 0.001; TPC, total phenolic content; TFC, total flavonoids content; ABTS, ABTS radical-scavenging activity; DPPH, DPPH radical-scavenging activity; FRAP, ferric reducing antioxidant power; RMA, rosmarinic acid; CARO, carnosol; CARA, carnosic acid; SLA, salvianolic acid.

## Data Availability

Data is contained within the article.

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
