# Peer review of "Bioactive Profile of Distilled Solid By-Products of Rosemary, Greek Sage and Spearmint as Affected by Distillation Methods"

_molecules, 2022, doi:10.3390/molecules27249058_

Round 1

Reviewer 1 Report

1.What is the significance of the difference between HD and SD or MAHD methods for this research?

2.Which compounds in Table 2 belong to flavonoids? Is the content of flavonoids higher than that of phenols?

3.peak 22/23/24/29 were characterized by the presence of rosmanol isomers. Why is it considered as isomer of Rosmanol?

4.There are still some small errors to check ,for example :(peaks 20, 21, 22 and 27), four peaks (8, 13 and 19)……

Author Response

Reviewer #1

We thank Reviewer #1 for his/her valuable comments.

Comment 1:

What is the significance of the difference between HD and SD or MAHD methods for this research?

Response:

According to Green Chemistry principles, conventional extraction and distillation processes are considered outdated and tend to be replaced by more environmentally friendly ones such as MAHD. Although the majority of commercially available essential oils are obtained through SD and/or HD, these methods present several disadvantages, namely high energy and water consumption, losses and thermal degradation of volatile compounds and long distillation times. Thus, the obtained results of this research pointed out that MAHD could be applied in the food, pharmacy and cosmetic industries in order to extract bioactive compounds with high quality and in addition contributing to circular economy.

Comment 2:

Which compounds in Table 2 belong to flavonoids? Is the content of flavonoids higher than that of phenols?

Response:

Regarding Table 2, flavonoids and flavonoid derivatives are: gallocatechin isomer, luteolin-7-O-rutinoside, luteolin-7-O-glucuronide and hesperidin. The contents of the above flavonoids are presented in Table 3, except gallocatechin isomer, due to the lack of standard compound. As you can see their contents is < 5% of the major compounds belonging to phenolic acids derivatives.

Comment 3:

peak 22/23/24/29 were characterized by the presence of rosmanol isomers. Why is it considered as isomer of Rosmanol?

Response:

Peaks 22, 23, 24, 27 and 29 have similar pseudo-molecular ion m/z 345 and were characterized as rosmanol isomers according to the literature data. Its isomers may be epiisorosmanol or epirosmanol or 12-O-methyl-carnosic acid. So, we correct the terms ‘rosmanol isomer’ as ‘rosmanol or isomer’.

(Relevant references:

  1. Zimmermann, B.F.; Walch, S.G.; Tinzoh, L.N.; Stühlinger, W.; Lachenmeier, D.W. Rapid UHPLC determination of polyphenols in aqueous infusions of Salvia officinalis L. (sage tea). J. Chrom. B 2011, 879, 2459–2464
  2. Montesino, N.L.; Kaiser, M.; Mäser, P.; Schmidt, T.J. Salvia officinalis L.: Antitrypanosomal Activity and Active Constituents against Trypanosoma brucei rhodesiense. Molecules 2021, 26, 322.)

Comment 4:

There are still some small errors to check, for example: (peaks 20, 21, 22 and 27), four peaks (8, 13 and 19)……

Response:

Thank you very much for this notification. We have corrected the number of peaks.

Reviewer 2 Report

The manuscript described the effects of different distilled methods (microwave-assisted hydro-distillation, hydro-distillation, steam-distillation) on total phenolic content, total flavonoid content, and antioxidant activity of solid by-products from Rosemary, Greek Sage, and Spearmint. Generally, it is well-written, but some parts need to improve. The specific comments are as follows:

Lines 75-77: As the authors stated microwave-assisted processes have been comprehensively applied for the extraction of bioactive compounds, what reason hinders its application to EOs distillation? Please explain more specifically. 

Lines 81-82: the authors stated that microwave-assisted distillation is an eco-friendly and innovative distillation method, but in Lines 81-82, it states that all the above distillation processes (including MAD) result in two different types of products, and the remaining solid biomass waste is no industrial value. Does it mean the by-products generated from MAD will be disposed of by the environment directly? In that case, I don't think MAD is an eco-friendly method. 

Lines 100-103: although the authors tried to show the novelty of their research here, the authors stated there is limited information, when you google on the scholar, there are a lot of information like "A novel approach for isolation of essential oil from fresh leaves of Magnolia sieboldii using microwave-assisted simultaneous distillation and extraction", "Solvent Free Microwave-Assisted Extraction of Antioxidants from Sea Buckthorn (Hippophae rhamnoides) Food By-Products" "Bioactive Compounds from Agricultural Residues, Their Obtaining Techniques, and the Antimicrobial Effect as Postharvest Additives" and so on. Please state the differences and novelty

Figure 2: it is better to add the table to list what the compositions are in number, it is a little bit messy list in the maintext. 

Tables: I am unsure what the treatment of raw?  Do you mean the control group? It is not clearly described how did the authors treat the raw?

Author Response

Reviewer #2

We thank Reviewer #2 for his/her comments.

Comment 1:

Lines 75-77: As the authors stated microwave-assisted processes have been comprehensively applied for the extraction of bioactive compounds, what reason hinders its application to EOs distillation? Please explain more specifically.

Response:

A more detailed explanation is presented in lines 78 – 81 of the revised manuscript according to Reviewer’s comment.

Comment 2:

Lines 81-82: the authors stated that microwave-assisted distillation is an eco-friendly and innovative distillation method, but in Lines 81-82, it states that all the above distillation processes (including MAD) result in two different types of products, and the remaining solid biomass waste is no industrial value. Does it mean the by-products generated from MAD will be disposed of by the environment directly? In that case, I don't think MAD is an eco-friendly method.

Response:

The remaining solid biomass has no important direct industrial value without further processing. By-products are discarded in the environment unless they are further processed for the recovery of bioactive compounds after distillation. Correction has been made in line 91 of the revised manuscript.

Regarding MAD, the sustainability of the method lies in the shorter extraction/distillation times required and consequently to the lower energy consumption during extraction/distillation, compared to conventional methods. No method can be completely eco-friendly, but MAD offers more advantages over conventional ones. However, even though MAD also generates by-products, as stated in the results (132 -140 of the revised manuscript), these by-products are rich in bioactive compounds and, in fact, have a higher phenolic content compared to the respective by-products from the other two distillation methods, due to milder process conditions.

Comment 3:

Lines 100-103: although the authors tried to show the novelty of their research here, the authors stated there is limited information, when you google on the scholar, there are a lot of information like "A novel approach for isolation of essential oil from fresh leaves of Magnolia sieboldii using microwave-assisted simultaneous distillation and extraction", "Solvent Free Microwave-Assisted Extraction of Antioxidants from Sea Buckthorn (Hippophae rhamnoides) Food By-Products" "Bioactive Compounds from Agricultural Residues, Their Obtaining Techniques, and the Antimicrobial Effect as Postharvest Additives" and so on. Please state the differences and novelty.

Response:

The aim of the present study was to investigate the effect of different distillation methods on the overall phenolic profile and composition of the respective solid waste in three major aromatic herbs of the Mediterranean region. Most studies focus on the recovery of bioactive compounds from by-products using green extraction techniques. The novelty of the present study is the comparison of the distillation methods (conventional vs green) on the bioactive composition of the by-products in order to highlight the important effect of the initial treatment of the raw material to the profile of the generated solid waste.

Comment 4:

Figure 2: it is better to add the table to list what the compositions are in number; it is a little bit messy list in the maintext.

Response:

The explanation of peak number has been added in the Fig. 2.

Comment 5:

Tables: I am unsure what the treatment of raw?  Do you mean the control group? It is not clearly described how did the authors treat the raw?

Response:

Raw is referred to the initial raw plant material, prior to essential oil distillation. Explanation has been added in lines 114-115 of the revised manuscript.

Reviewer 3 Report

The manuscript describes novel approach to the extraction of important compound in the context of green chemistry.

Overall, the manuscript is well written. Results are well structured and disccussion is sound and logical.

Author Response

Reviewer #3

We thank Reviewer #3 for his/her comments.

Comment 1:

The manuscript describes novel approach to the extraction of important compound in the context of green chemistry.

Overall, the manuscript is well written. Results are well structured and discussion is sound and logical.

Round 2

Reviewer 2 Report

The authors have addressed my comments